# A Generalized and Modular Framework for Digital Generation of Composite Microstructures

**Ahmet Cecen [1], Berkay Yucel [2] and Surya R. Kalidindi [1,2,]***

[1] School of Computational Science and Engineering, Georgia Tech, Atlanta, GA 30332, USA; ahmetcecen@gatech.edu
[2] Woodruff School of Mechanical Engineering, Georgia Tech, Atlanta, GA 30332, USA; beyucel@gatech.edu
[*] Correspondence: surya.kalidindi@me.gatech.edu

**Abstract:** This paper presents a generalized framework for the digital generation of composite microstructures using filter-based approaches that can devise and utilize a wide variety of cost functions reflecting the desired targets on geometrical and statistical measures. The use of filter-based approaches leads to remarkable computational advantages compared to the conventional approaches used currently for microstructure generation. The framework provides a highly modular and flexible approach to generate stochastic ensembles of microstructures meeting user-defined microstructural characteristics. The proposed framework is illustrated in this paper through selected case studies.

**Keywords:** composite microstructures; microstructure generators; microstructure statistics; image filters





## 1. Introduction

Composite materials represent an important class of advanced materials due to their superior properties for many application domains [1–3]. Digital generation of a composite material's internal structure (often simply referred as microstructure) plays an important role in advancing our understanding of the quantitative connections between the microstructure and its associated properties, as well as those between the processing history and the final microstructure attained in a manufactured part. Indeed, a digital microstructure is often the input into most commonly employed multiscale material simulation toolsets (e.g., micromechanical finite element models [4–6] and phase-field models [7,8]). Recent efforts in the generation of digital microstructures have focused on problems such as the generation of a statistically similar ensemble of microstructures given a limited amount of available (reference) experimental observations [9–12], the generation of representative three-dimensional (3D) volume elements of material microstructures based on statistics gathered from two-dimensional (2D) microscopy scans or images from a sample [13–21], and the visualization of microstructures corresponding to a prescribed set of measured or predicted spatial correlations [19,22–28].

Broadly speaking, the generation of digital microstructures involves two main steps: (i) selection and prioritization of the metrics, statistics, and/or geometric constraints targeted in the microstructures to be generated, and (ii) development and deployment of computational algorithms for efficiently arriving at digital instantiations of microstructures meeting these criteria. Generation targets are most commonly formulated as shape and size distributions of the required salient features in the microstructure. These could be specified in the form of idealized geometries (e.g., spheres, cylinders, ellipses) that could be placed in a given representative volume element of the microstructure [14] with constraints on separation or overlap [29]. Alternately, the target statistics can also be specified as a suitable set of spatial correlations (e.g., n-point spatial correlations [30,31]). The most commonly used approaches for digital generation of microstructures can be classified as "packing" algorithms. In these approaches, one aims to pack selected geometric shapes

(e.g., spheres and cylinders) in a given representative volume (or area), usually in a non-overlapping arrangement, while reaching the prescribed statistics or metrics (generally expressed as volume fractions or size and shape distributions of the selected geometric shapes). Exemplary among these efforts is the work of Tschopp et al. [29], where one picks a point in the microstructure randomly and checks if the placement of the selected feature at the selected point produces an overlap with the pre-existing features in the microstructure being generated. If an overlap exists, the point is discarded. If there is no overlap, the selected feature is placed at the selected point in the microstructure. This process is then repeated until one arrives as close as possible to the target statistics. While this strategy can be employed effectively for a broad range of multiphase composite microstructures that conform to the idealizations and assumptions made regarding structural geometry, various deficiencies can immediately be noticed with respect to scalability. As the volume fractions and sizes of the microstructural features to be placed increase, the algorithm will result in a quickly exploding number of placement failures (i.e., it becomes increasingly hard to find a location with no overlap); the problem becomes even more challenging for generations of large 3D microstructures. The dependence on random trials makes it difficult to target specific statistics (i.e., distributions) on the separation distances (negative distances can be treated as overlap) between the features or their orientations (for non-equiaxed features).

Polycrystalline microstructures can be considered a special class of composite microstructures, where the microscale constituents are grains of different crystal lattice orientations. For such microstructures, one typically employs a strategy that results in space-filling arrangements of the grains. The strategies explored in the literature have included Voronoi tessellations (e.g., [32]) and dilations of elliptical seeds [13,33]. Once a tessellated volume is generated (usually targeting prescribed distributions on grain size, grain shape, and/or the number of neighbors), the problem of generating a digital polycrystalline microstructure reduces to the assignment of local states or characteristics (i.e., orientation and phase identifier) to each seed (i.e., voronoi cells or grains), while achieving desired statistics of the phase volume fractions, crystal orientations, and/or misorientations. In general, the two steps described above are pursued in a completely uncoupled manner, i.e., there is usually no attempt to further improve the shape and size distributions of specific grain orientations in the second step.

A major deficiency of the techniques described above is that they are not easily extended to situations in which one desires to simultaneously target a large number of different microstructure statistics. Furthermore, these methods cannot handle multiple geometrical or statistical constraints simultaneously in a computationally efficient manner, especially since they were not designed with generalization in mind. As such, most approaches used in the current literature will yield good results only for the specific types of microstructures and geometries they were designed to address. Furthermore, many of the approaches described above are generally not able to reproduce the complex features observed in experimentally documented microstructures.

In recent years, there have been various attempts to address the limitations described above using new approaches to microstructure generation based on algorithms from the field of texture synthesis [34]. The word texture, in this context, generally refers to a digital image with a distinct and visually identifiable pattern/order, usually consisting of repetitive elements. The objective is, then, to generate a larger image or volume showing a similar visual pattern or order in a seamless and aesthetically appealing way. This can be achieved, for instance, by considering the histogram of the color/grayscale values on a reference texture as a descriptive metric, and generating larger images (or, equivalently, textures) that match the target histogram. These methods can be extended to include the volume fraction of constituents or orientation distributions. Alternatively, methods of neighborhood matching can be utilized, where the immediate vicinity of each pixel in the generated texture is guided by trends sampled from the reference texture. Among these efforts, Markov fandom fields (MRF)-based synthesis has found, by far, the most common usage in the materials domain [35]. In the most exemplary formulation of this approach

presented by Kopf et al. [35], concepts of volume fraction matching and MRF-based neighborhood matching were combined in an optimization loop, which resulted in the utilization of both local (neighbors) and global (volume fractions) information during the generation process. While this method shows tremendous promise, it inherently assumes that the state assigned to each pixel in a microstructure is only determined by the pixels in its close neighborhood. Although it is clear from these implementations that the simultaneous consideration of both the global statistical measures and neighborhood features is important for the accuracy and the visual quality of the generated microstructures [15], this approach is not practical when targeting medium or long range structural order, as the computational cost increases dramatically with the consideration of larger neighborhood sizes.

The inadequacy of texture synthesis approaches for microstructure generation can be understood by recognizing that texture synthesis is mainly concerned with the generation of a smooth and visually appealing image from a given reference. Texture synthesis methods utilize exclusively the information from a sample or multiple samples and create variations. On the other hand, the goal of microstructure generation is to create instances that could have come from the population sampled. Texture synthesis methods focus heavily on visual similarity, which results in the over-utilization of absolute information from the reference, and minimal or weak utilization of the rigorous spatial statistics that could be extracted from the reference. In other words, in microstructure generation, we want to emphasize the matching of a large number of robust statistical measures, even if the structure looks less similar visually, as our interest is in mimicking the overall properties exhibited by the composite material. In this context, it is important to understand and acknowledge the important role of spatial correlations in controlling the properties associated with material microstructures [36,37].

A completely different approach to the generation of digital microstructures comes from formulating the problem as a minimization of the difference between the targeted statistics and the corresponding statistics for the generated microstructures, and solving this problem using optimization toolsets. For a meaningful application of this approach, one needs to select a suitably large set of target statistics. This is because, with a small set of statistics, the number of potential solutions is quite large. The complete set of two-point spatial correlations has been targeted in some recent microstructure generation efforts [27,38]. Since these approaches employ optimization strategies, they implicitly assume that there is an underlying absolute target to be matched. In other words, similar to the case of texture synthesis, these methods target the reference strongly, and only weakly target the population from which the reference was sampled. Furthermore, most phase recovery approaches [27,38] can be applied only when the complete set of two-point spatial correlations are available.

This paper aims to bring together all the seemingly disparate approaches described above in a single consistent and modular framework. This new overarching framework allows seamless interplay of all the different microstructure generation strategies described above from the current literature. Assembled specifically with microstructure instantiation in mind, the framework can accommodate both geometrical (i.e., overlap constraint and distance constraint) and statistical constraints (two-point auto correlations) within user-defined tolerances. The framework allows the specification of arbitrary shaped features (could come directly from experimental microstructures) and is designed to be fully compatible with any of the existing microstructure generators. This new framework is first described in the next section, and subsequently demonstrated with multiple case studies.

## 2. Microstructure Generation Framework

As previously mentioned, the goal of microstructure generation is to meet a specified set of target statistics within user-defined tolerances. This targeted set of statistics is considered to be rank-deficient, in the sense that multiple microstructure instantiations are expected to meet them within the set tolerances. Therefore, the problem, here, is formulated as microstructure generation that aims to minimize a suitably defined error between the

statistics of the generated microstructure and the targeted set. This approach allows one to think of the generated microstructures as statistical volume elements (SVEs) [39,40]. In contrast, one might be interested in generating representative volume elements (RVEs) corresponding to a clearly specified set of targeted microstructure statistics. In order to arrive at RVEs, one can try to select specific SVEs (from a large set of generated SVEs) whose average statistics match the target statistics to arbitrary (user specified) precision. However, this approach is likely to be inefficient and may need a very large set of generated SVEs. Alternately, it would be much more efficient to select a set of weighted SVEs (i.e., WSVEs [41–43]) whose selection and weights are optimized to match the weight-averaged statistics of the SVEs to the target statistics to arbitrary precision.

The generalized microstructure generation framework presented in this paper employs the iterative approach presented schematically in Figure 1. The generation starts with a trial (guess) of the microstructure to be instantiated. This could be taken from a compiled library of potential microstructures or generated using random (Gaussian) noise and a suitable segmentation [44]. The framework comprises the following main steps: (1) the creation and/or updating of a library of features of interest using available reference structures/images, (2) selection of a specific feature for placement, (3) efficient computation of the cost functions describing the improvements in all the prescribed geometric and statistical distributions for all possible placement locations of the selected feature in the microstructure being generated, (4) determination of the placement location of the selected feature employing suitable heuristics, and (5) an evaluation of how the updated microstructure meets all the prescribed criteria. If the prescribed criteria are not met, one might proceed to the next iteration by starting with either step 1 (i.e., updating the feature library) or with step 2 (i.e., selection of a different feature from the library). If all the prescribed criteria are met, the iterations stop, producing a potential microstructure instantiation of interest. Additional instantiations can be produced using different initial guesses (i.e., different initial microstructures). Note that the geometrical constraints (target overlap and distance constraints) and statistical constraints (target two-point spatial correlations) are treated as user-defined parameters in the presented framework. Therefore, the framework allows for the generation of a wide variety of microstructures at low computational costs. Details of each step identified above are further elaborated in the next sections.

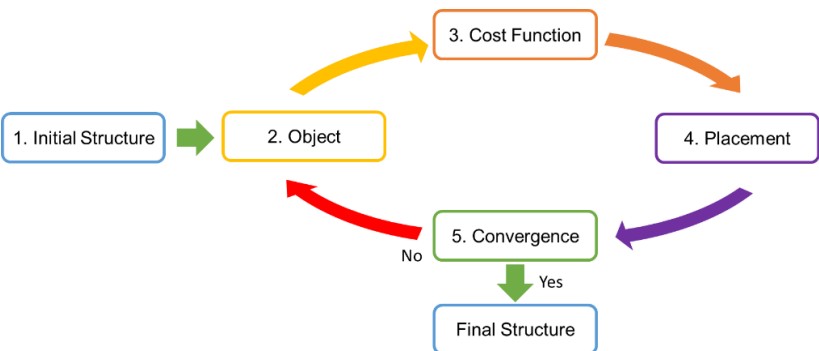

**Figure 1.** The main workflow of the microstructure generation framework comprising the following steps: (1) creation of an initial guess of the microstructure, (2) selection of the object to be placed from a library of candidate objects, (3) computation of the geometric and statistical cost functions, (4) placement of the object at an optimum location based on heuristics, and (5) assessment of convergence.

### 2.1. Feature Library

The overall strategy employed here relies on the identification and placement of features in the microstructure volume meeting the prescribed statistics. As such, the generation of a suitable library of potential features is an important component of this process. Generally, there are two strategies to accomplish this task. The first approach relies on employing standard simple geometries that may be combined or merged suitably to produce more complex features. For example, in generating 2D microstructures, one

might consider a broad selection of rectangles, circles, and triangles of different shapes and sizes as the initial simple features. Similarly, in generating 3D microstructures, one might consider a broad selection of rectangular parallelepipeds, spheres, cylinders, and pyramids for the simple initial features. One can then employ the algorithms described in this work to allow for the generation of compounded features by overlaying individual features (through the iterative process depicted in Figure 1). Consequently, the overall set of distinct features that can be found in the final generated microstructure is significantly richer than those in the feature library.

The second approach to creating a feature library relies on extracting the features directly from the reference images. Here again, one has two choices. As a first option, one can custom-select features from the reference images by cutting out specific portions of arbitrary sizes and shapes. In other words, the features can be irregularly shaped and/or disjointed cut-outs from the reference images that capture the salient features desired in the microstructure to be generated. As a second option, instead of custom-selecting the features, one can also randomly select small portions of the reference images. The selection of the sizes and shapes of these smaller images can also be randomized.

While it is tempting to accumulate a large library of features, one should recognize that the computational cost scales with the number of distinct features included in the generation process. Therefore, one should exercise prudence in selecting the features of interest. One can also tweak the feature library between the iterations, i.e., remove features that do not seem to be of value in achieving the desired statistics and add new ones.

### 2.2. Cost Functions

In the context of the present work, cost functions quantify the benefits (or penalties) for placing a selected feature at a selected location in the generated microstructure in efforts to move the microstructure statistics closer to the targeted statistics (including geometric constraints). The central impediment in evaluating these cost functions is their high computational cost. If one considers the generation of a microstructure with $S$ spatial bins (or voxels), then one needs to evaluate the cost function $S$ times to decide on the optimal placement of the selected feature. One of the main contributions of this paper is to demonstrate the computational advantages of using filters in evaluating these cost functions. More specifically, it will be demonstrated that a broad variety of microstructures with very tight and specific constraints can be generated entirely with $O(S\log S)$ computational complexity using filters [45]. Furthermore, the computational cost with this approach is independent of the complexity of the image or the type and size of the feature.

Although a broad variety of cost functions are possible, they can be classified into two major groups in the context of microstructure generation. The first group is aimed at geometrical constraints, while the second targets user-specified spatial correlations. These are described in detail next.

### 2.2.1. Overlap and Distance Cost Functions

The cost functions associated with meeting most geometrical constraints can be formulated as an overlap cost function (OCF). As the name suggests, this function quantifies the degree of overlap between a new object to be placed in a microstructure and the existing objects already present in the microstructure. The most common practice in current literature for evaluating the OCF is often a brute force approach involving random placements and repeated trials [29]. Such approaches typically result in a computational cost that is bounded, roughly, at $O(S^2)$. Consequently, the brute force approach becomes impractical for both large microstructure domains and large objects. Moreover, there is currently no formal approach to allow partial overlaps with specified statistics (which may include specification of desired locations of overlaps).

In this work, we propose addressing these cost functions using a filter-based approach that takes advantage of the convolution properties of discrete Fourier transforms (DFTs) computed with a cost of $O(S\log S)$ using the fast Fourier transform (FFT) algorithm [46].

In addition to the low computational cost, the other main advantages of this approach include: (i) ability to place objects at multiple locations in a single iteration, especially in the early stages of microstructure generation, and (ii) flexibility for allowing partial overlaps or gaps between objects with specified statistics.

The proposed filter-based approach is illustrated in Figure 2 with a simple example of an iteration in the generation of a 2D microstructure (can be trivially extended to 3D microstructures). The initial microstructure (i.e., at the start of an iteration) is shown here with differently oriented black rectangular objects placed on a white background. The black border on the image is not a part of the microstructure and has only been added to improve visualization (to denote the edges of the image). The object to be placed is a specifically oriented rectangle, also shown as an image of the exact same size as the microstructure. These images are mathematically represented as arrays, where the black pixels are assigned a value of one and the white pixels are assigned a value of zero. A simple convolution of the object image with the microstructure image normalized by the object size (i.e., number of pixel/voxels in the object), produces the desired OCF (also shown in Figure 2). Mathematically, this computation can be expressed as

$$\text{OCF} = \frac{\Im^{-1}\big(\Im(\text{Microstructure})^* \odot \Im(\text{Object})\ \big)}{\text{ObjectSize}} \tag{1}$$

where $\Im(\ )$ denotes a DFT operator (implemented using an FFT algorithm), $\odot$ denotes a simple element-wise multiplication of the corresponding frequency terms (i.e., a Hadamard product), and the superscript * denotes the complex conjugate. One of the consequences of employing DFTs is that they implicitly impose periodicity conditions across the image boundaries. In many instances, this is an advantage. For example, in most numerical simulations of microscale materials phenomena, one routinely imposes periodic boundary conditions on the representative volume element of the material microstructure. However, in cases where one desires to avoid this assumption, one can employ a variety of padding strategies (cf. [47]) to get around this limitation. Each pixel (voxel) value in the OCF map shown in Figure 2 describes the percent overlap of the new object with previously existing objects, if the new object were to be placed centered on that pixel (voxel).

As mentioned earlier, we desire versatility in imposing additional geometric constraints and/or statistics in the placement of objects (e.g., controlling the gaps between objects). It is proposed that we provide this versatility by adapting a common concept from the image processing field called distance transform [48–50] and computing a distance cost function (DCF) defined as

$$\text{DCF}_s^+ = \min_{s'}\text{dist}(s, s'),\ \ s \in \text{OCF} = 0\%,\ \ s' \in \text{OCF} > 0\% \tag{2}$$

$$\text{DCF}_s^- = \min_{s'}\text{dist}(s, s'),\ \ s \in \text{OCF} > 0\%,\ \ s' \in \text{OCF} = 0\% \tag{3}$$

$$\text{DCF} = \text{DCF}_s^+ - \text{DCF}_s^- \tag{4}$$

where $s$ and $s'$ index the voxels in the microstructure and the dist$()$ function outputs the Euclidean distance, i.e., dist$(s, s') = s - s'$. It is also convenient to think of the indices $s$ and $s'$ as integer array indices. For example, for a 3D microstructure, one could express $s = \{s_1, s_2, s_3\}$, with $S_i$ taking integer values. Equation (2) defines a distance for the voxels without any overlap (based on the computed OCF) to the nearest pre-existing object in the microstructure; this function provides a measure of the gap between the object to be placed and the existing object. Similarly, Equation (3) defines an overlap distance for the voxels with overlap (based on the computed OCF). Equation (4) combines both these measures in a single signed DCF. The computations described in Equations (2) and (3) can be performed effectively using MATLAB's bwdist [48,51] function with O(S) complexity.

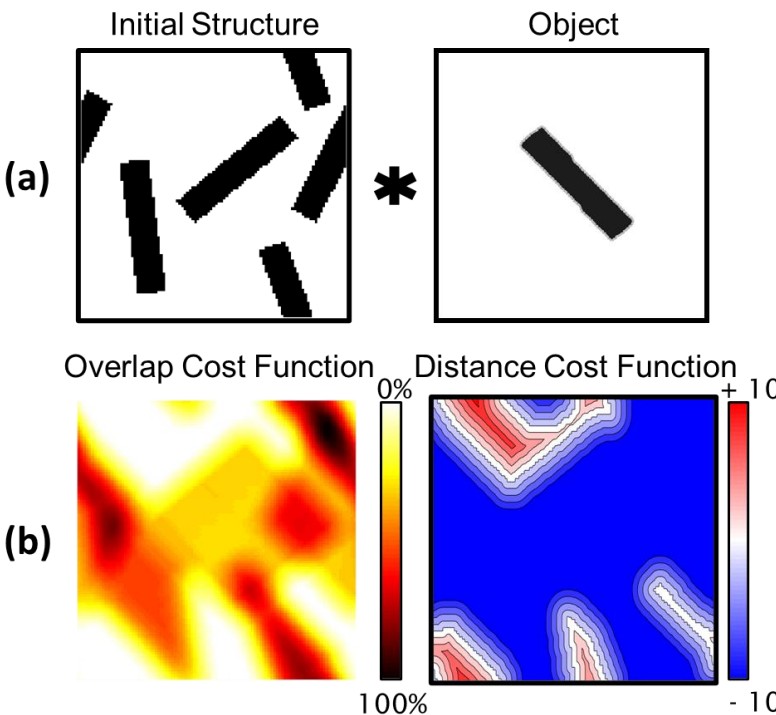

**Figure 2.** (**a**) Initial structure before object placement (left) along with the object to be placed (right). (**b**) Illustration of the computation of the overlap cost function (OCF) and the distance cost function (DCF).

The framework and concepts introduced above offer tremendous flexibility for many potential extensions. For example, the microstructure in Figure 2 had two local states (these refer to the distinct microscale constituents present in the microstructure) that were colored white and black. Although the example above discussed the overlap cost function for the placement of a selected object with respect to pre-existing black objects, it should be easy to see that the cost functions can be defined separately for each local state present in the microstructure. Therefore, for the general case of multiple local states, one can define OCF[$h$] as the overlap cost function for the placement of a selected object with respect to pre-existing objects belonging to local state $h$. Similarly, one can, then, extend the definition of DCF to identify the distances (i.e., gaps and overlaps) separately with each local state, expressed as DCF[$h$]. As further examples, it is also possible to use measures of distances other than the Euclidean distance (such as the Chebyshev distance [52] or the quasi-Euclidean distance [53]) in defining the DCF.

It is pointed out that a very wide variety of statistics on the overlaps and gaps between objects can be accommodated in the microstructure generation simply by using the OCF and DCF in conjunction with each other. Figure 3 depicts examples of the different compounded object configurations produced by imposing different combinations of OCF and DCF. In these examples, the blue object represents a pre-existing object, while the green object represents a new object to be added to the microstructure. In general, it is our experience that one obtains better control on the placement of microstructural objects by using both OCF and DCF. Using only one of the cost functions generally results in unintended object morphologies.

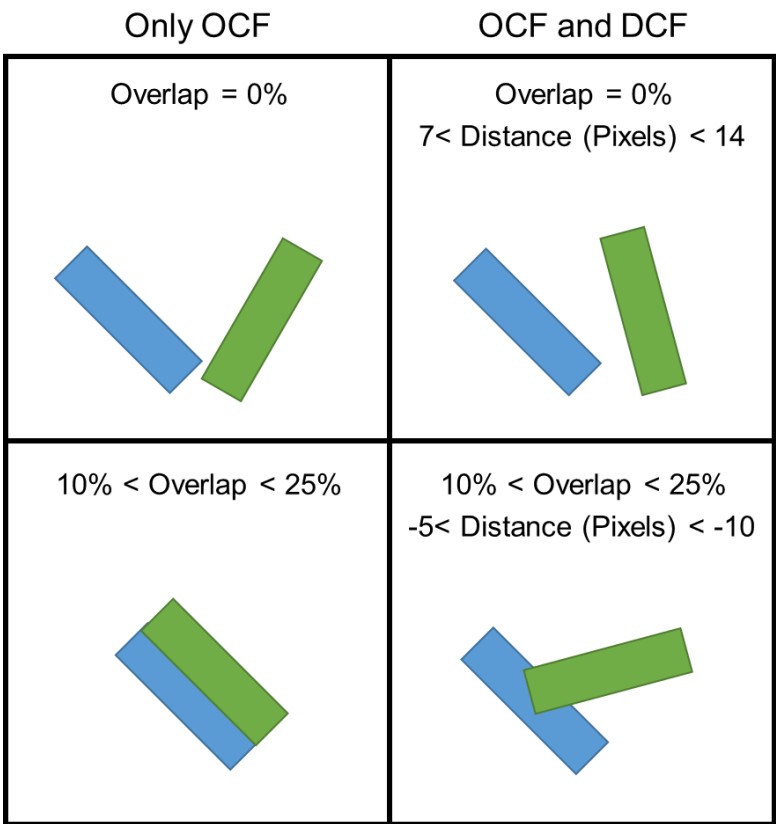

**Figure 3.** Examples illustrating various placement modes that can be achieved using an OCF and a DCF in conjunction. For each example, the blue rectangle is already in the microstructure, and the green rectangle is to be placed.

Figure 3 illustrates only the simplest of the compounded geometrical configurations controlled with the direct application of the OCF and DCF presented earlier. One can be much more creative in the application of these cost functions. For example, one can define and employ meta-local states in the objects. As a simple example, in the rectangular object we have been considering thus far, one might want to differentiate the end regions from the middle portions. For this purpose, the portions of the object to be added can be labelled distinctly. As an example, the object to be added is shown with yellow ends and a green middle portion in Figure 4. In other words, the pixels in these regions have been assigned different local states, referred here as meta-local states. In a similar way, the pre-existing object in the microstructure can also be assigned different metal-local states (see the yellow and blue colored regions in the pre-existing object in Figure 4). The creation of these meta-local states offers tremendous control and specificity in adding desired geometrical configurations to the generated microstructure. This capability is illustrated in Figure 4 by chaining multiple overlap and distance-based constraints in a sequential manner. In other words, as one applies additional constraints in a chained (sequential) manner, one can ensure that only the desired geometrical configurations are successfully incorporated into the generated microstructure. In the example shown, the objective was to make sure the objects connect only on the ends at wide angles to each other (without significant overlap).

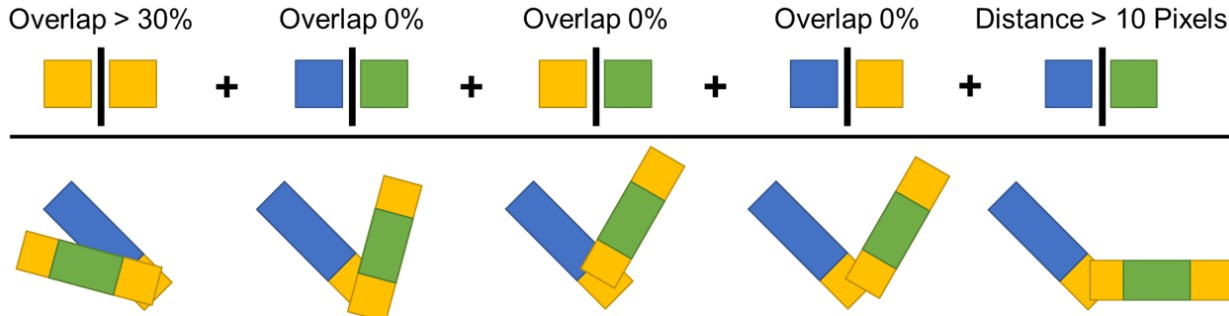

**Figure 4.** Examples showing how the placement mode evolves as four OCFs and a DCF are chained to achieve a specific criteria. For each example, the blue–yellow rectangle is already in the microstructure, and the yellow–green–yellow rectangle is to be placed. Yellow areas are labelled with meta-local states to correspond to the ends of the rectangle. The final goal is to achieve a placement where rectangles overlap exclusively in the yellow regions, while having a wide angle between them.

### 2.2.2. Two-Point Statistics-Based Cost Functions

As mentioned earlier, a number of microstructure generation efforts are likely to target specific sets of spatial correlations due to their important role in controlling the effective properties exhibited by the material [47,54–57]. Although these statistics are very different in how they are defined compared to the statistics on the geometrical constraints considered thus far, we will explore if the same toolsets described above can also be employed in spite of these differences.

One way to approach the task at hand is to define the object of interest as a single voxel in the microstructure. We, therefore, would like to define the cost function, quantifying the degree to which changing the local state in any selected voxel in the microstructure helps drive the microstructure towards targeted spatial correlations. We will also limit our attention, initially, to two-point spatial correlations in a two-phase composite, where a single autocorrelation is adequate to capture the complete set of two-point spatial correlations [26,27,47]. Furthermore, as discussed earlier, we will impose periodic boundary conditions on the microstructure.

We start by considering the changes to the autocorrelation when one changes the local state in one selected voxel in the microstructure. This is illustrated through the example shown in Figure 5. The example microstructure and autocorrelation computed using previously established algorithms [47] is shown as Figure 5a,b, respectively. In this example, the voxels colored yellow are assigned values of one, and voxels colored blue are assigned zero values. The autocorrelation in Figure 5b represents the yellow–yellow autocorrelation. Next, we identify a specific blue voxel (shown in red) and wish to quantify the changes in the autocorrelation when this voxel is flipped to a yellow voxel. It is important to recognize that the autocorrelation is simply the number of successes of finding yellow voxels at the head and tail of vectors placed into the microstructure normalized by the number of trials. In Figure 5b, all vectors are visualized as emanating from the center of the map. In other words, the center voxel in Figure 5b corresponds to a zero vector. Given this interpretation, it is easy to see that the additional vector counts created by flipping the voxel are easily visualized by moving the voxel to the center of the microstructure, while taking advantage of the periodicity assumptions, as shown in Figure 5c. Therefore, the corrections of the autocorrelation are simply, as defined by Figure 5c, normalized by the number of spatial bins in the microstructure. It is important to note that each correction made in Figure 5c needs to be made both for the vector itself as well as its negative (autocorrelations are symmetric with respect to center).

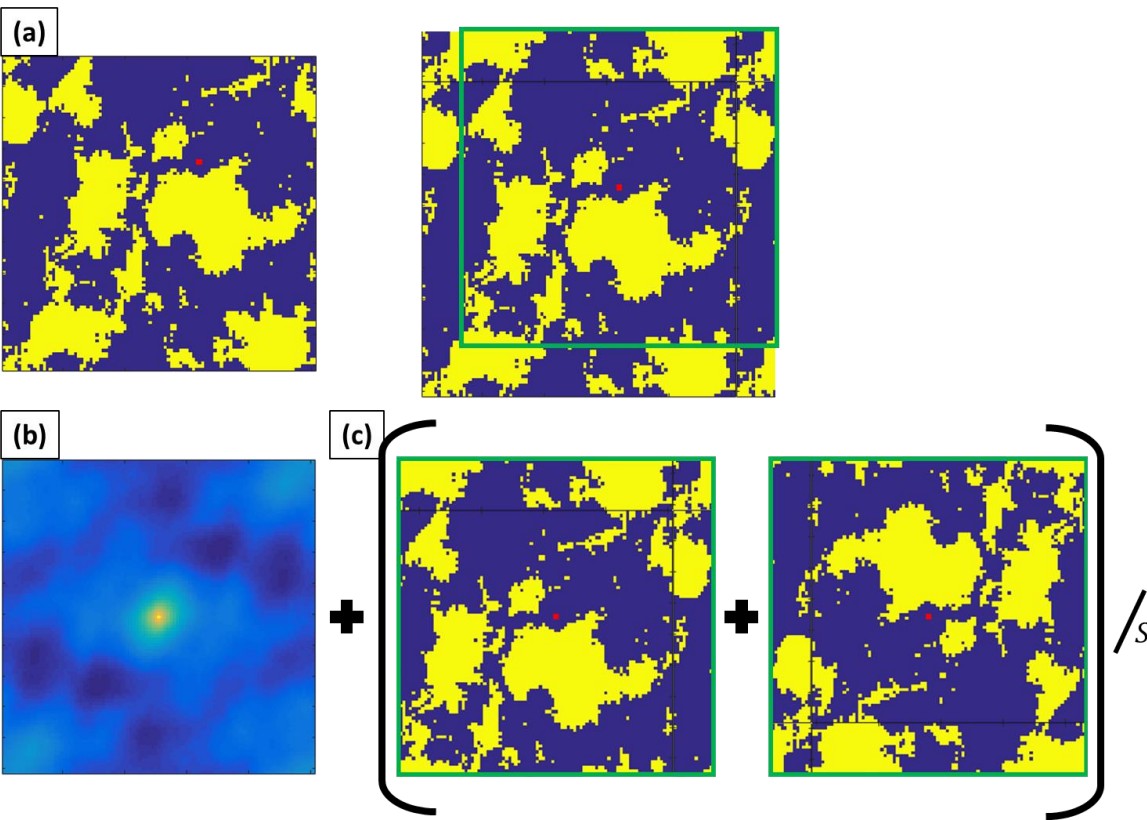

**Figure 5.** Illustration of the update of the autocorrelations with the change in the local state of a single pixel. (**a**) The initial microstructure with the pixel to be changed shown in red. (**b**) Yellow–yellow autocorrelation. (**c**) The correction of the autocorrelation as a result of the change made to the red pixel.

In order to make the generation process target a particular set of spatial statistics, it is necessary to quantify how much closer the spatial statistics of the current microstructure will be to the target statistics if a particular pixel is filled with a particular phase. Let us assume that we have a microstructure that only needs a single pixel modification to reach the target two-point statistics. In this case, by applying the concept from Figure 5, we can test the effect of modifying each pixel until we find the right one (see Figure 6a). However, another way of achieving the same effect is, instead, to find the difference between the target and current statistics to establish an approximation of the desired neighborhood, and then search the microstructure for this neighborhood (see Figure 6b). While direct application of both concepts will yield a search of $O(S^2)$ complexity, the workflow described in Figure 6b can be realized by convolving the desired neighborhood with the current microstructure, resulting in an $O(SlogS)$ complexity.

The concept described above can be generalized for microstructures that require major modifications to reach the desired two-point statistics. For this purpose, the difference between the target statistics and the statistics of the current microstructure would be treated as a weighted filter. In this filter, larger weights would signify high priority neighborhoods. As such, modifications to the structure should prioritize pixels that contain a large quantity of high priority neighborhoods. Such a filter will be referred henceforth as the placement gain filter (PGF).

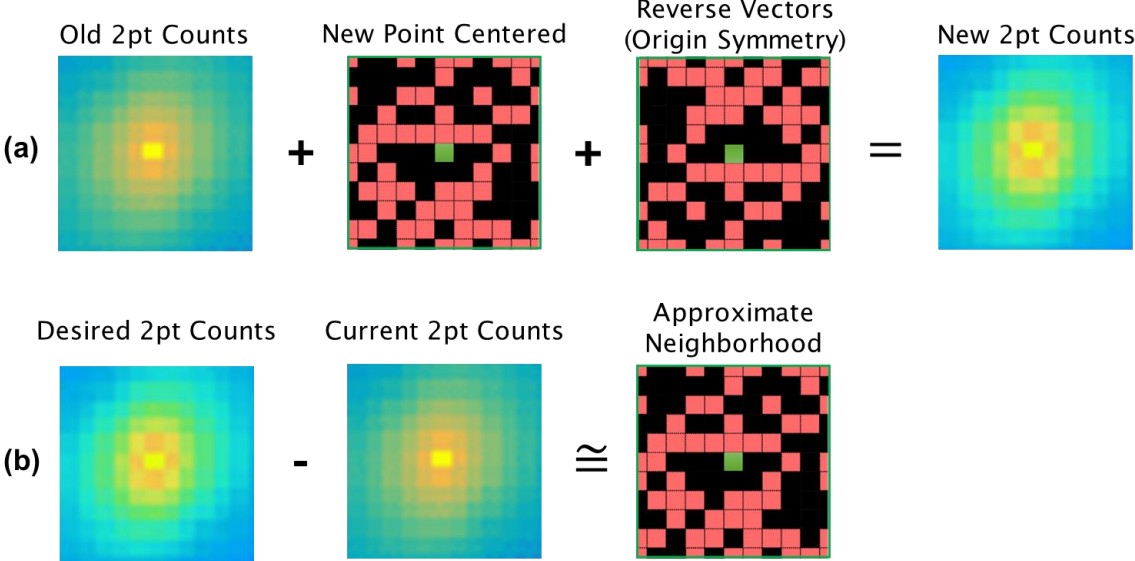

**Figure 6.** (**a**) The update concept for two-point statistics is shown graphically. (**b**) The graphical equation is rearranged to yield an approximation of the ideal neighborhood a pixel to be placed should possess.

Next, we demonstrate the calculation and utilization of the PGF for the placement of a selected object in an existing microstructure. Figure 7 shows an initial structure and the object to be placed, as well as both of their two-point autocorrelation plots. Figure 8 pictorially outlines the necessary operations to obtain the PGF for this object in the initial microstructure. Mathematically, this chained operation can be expressed as

$$\text{PGF} = \left( \overline{f} - \Im^{-1}\left( \Im(Image)^* \odot \Im(Image) \right) - \Im^{-1}\left( \Im(Object)^* \odot \Im(Object) \right) \right) \odot W \quad (5)$$

where $\overline{f}$ is the target two-point statistics and $W$ is the matrix of neighborhood weights. $W$ enables the generation process to prioritize a specific subset of the statistics, such as vectors of a particular length or orientation, based on physics/manufacturing requirements (for example, the statistics corresponding to long vectors have less impact on the properties compared to those corresponding to short vectors). The subtraction of the object two-point statistics from the target assumes that the object will be placed without overlap; however, the error introduced is minimal in most overlapping cases. Calculation of PGF only contains element-wise products, subtractions, and Fourier transforms for an O(SlogS) complexity.

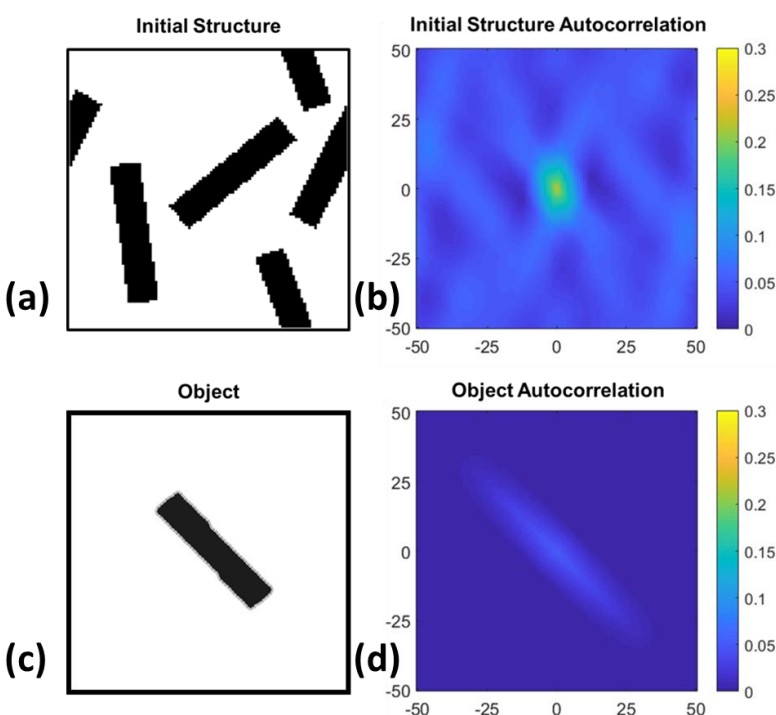

**Figure 7.** An example computation of the statistical cost function. (**a**) Initial structure before object placement. (**b**) Autocorrelation map of the initial structure. (**c**) Object to be placed. (**d**) Autocorrelation map of the object.

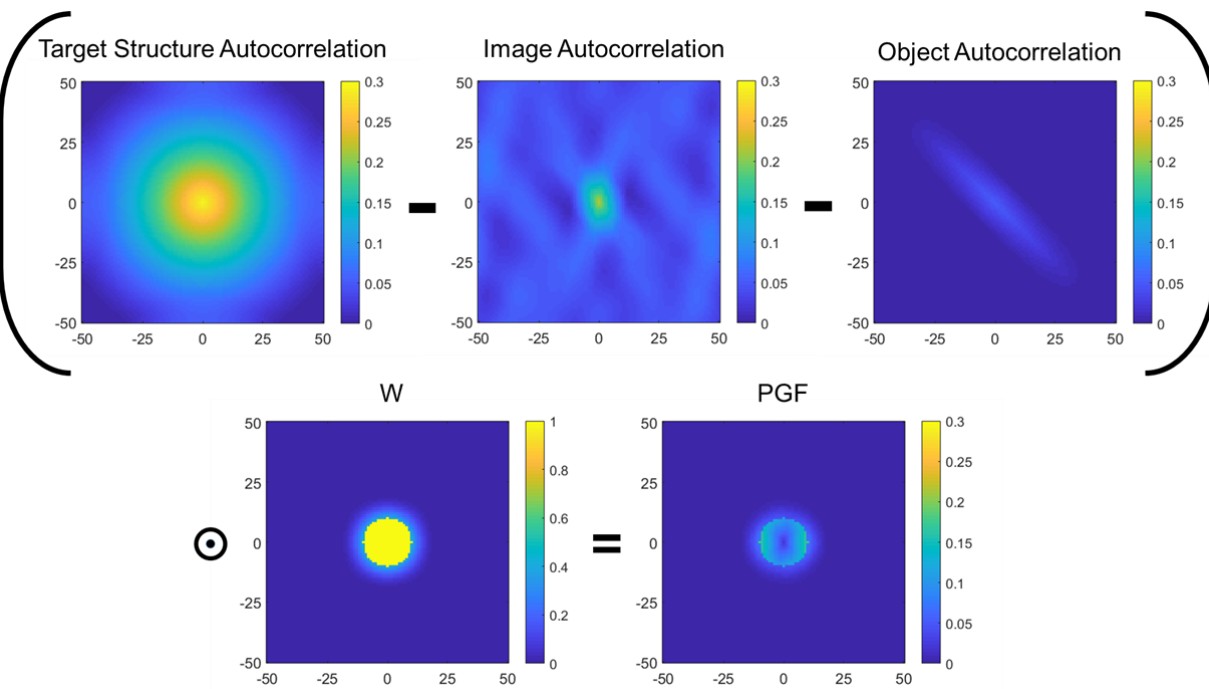

**Figure 8.** Graphical depiction of the operations involved in the calculation of the PFG (see Equation (7)).

In order to utilize the obtained PGF to calculate the statistical cost function (cost of placing an object towards satisfying the two-point statistics requirements), two more convolutions and a n element-wise multiplication are necessary; thus, the Fourier transform remains the operation with the highest complexity. SCF can be computed as

$$SCF = \Im^{-1}\left(\Im\left(\Im^{-1}(\Im(Image)^* \odot \Im(\text{PGF}))\odot Image\right)^* \odot \Im(Object)\right) \quad (6)$$

This operation is pictorially shown in Figure 9 for clarity. The resulting SCF can be used similarly to OCF and DCF to guide the placement of a new object.

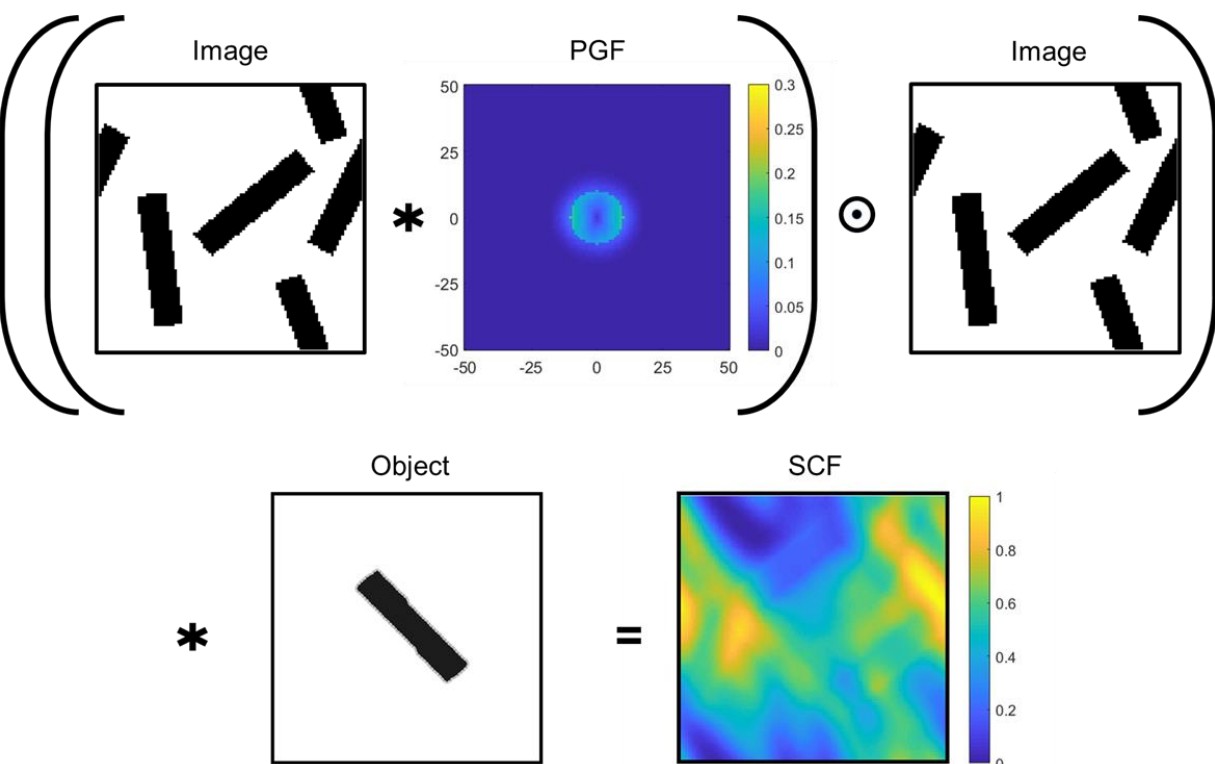

**Figure 9.** Graphical depiction of the operations involved in the calculation of the SCF (see Equation (8)).

### 2.3. Object Placement

The next step in any given generation iteration, after the computation of the cost function, is the actual placement of the object. There are two key steps to this decision. The first one involves the merging of information from the OCF, DCF, and SCF (see Figure 10). This can be achieved in various ways, such as addition (soft constraints), multiplication (hard constraints), or weighted independent voting. The examples presented in this work are generated using multiplicative joining:

$$Joint\ Cost\ Function = \text{OCF} \odot \text{DCF} \odot \text{SCF} \tag{7}$$

Note that the joint cost function can involve multiple OCFs, DCFs, and SCFs, corresponding to different local states. The second stage involves the selection of one or more points to place the object according to the joint cost function. To the very ill-posed nature of most microstructure generation problems, it is usually desirable to allow for random variation in the selection process so that the generation does not directly converge to the nearest local minima. The examples here treat the normalized joint cost function as a probability distribution and draw a sample for placement.

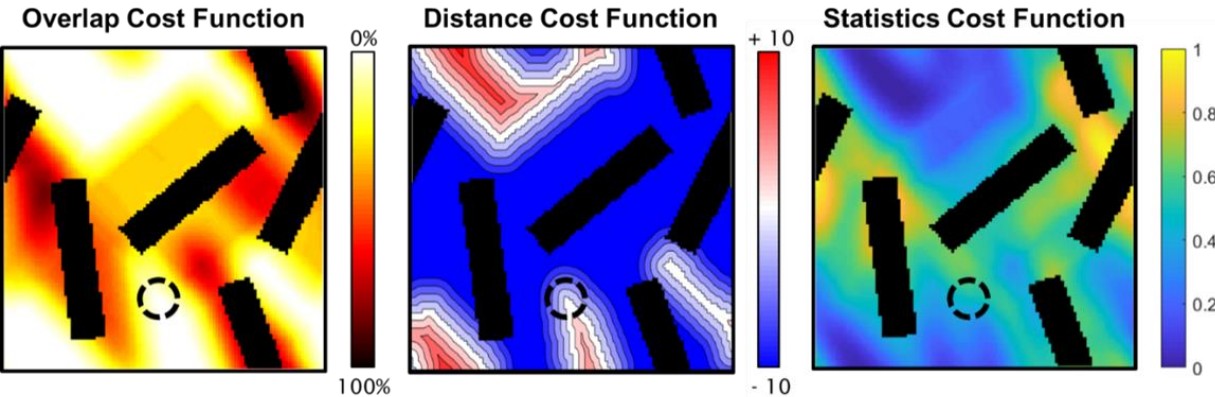

**Figure 10.** Computed cost functions for the generation example in this study.

### 2.4. Convergence

The final consideration in the microstructure generation framework is the stopping criteria. Either one-point (equivalent to volume fraction or number of objects) or two-point statistics convergence can be used to terminate the generation process. As with placement heuristics, error metrics and convergence methods are abundant in optimization literature [58]. For the examples presented in this paper, the following error metric is used and minimized when statistical criteria are imposed:

$$error = \frac{\left\| \left( \overline{f} - f \right) \odot W \right\|}{\left\| \overline{f} \odot W \right\|} \tag{8}$$

where $f$ is the two-point statistics of the current iteration of the generated microstructure, and W is the neighborhood weight matrix from the PGF calculation.

### 3. Case Studies

The framework described above is demonstrated using two case studies: generation of example digital microstructure images using arbitrary shapes and constraints demonstrating versatility, and generation of statistically similar alternatives to an experimentally obtained microstructure using statistical constraints.

### 3.1. Digital Generation of Microstructure Library

Figure 11 shows six microstructures generated using the protocols described in this paper. All the examples shown were generated at O(SlogS) computational cost. Figure 11a was generated using non-overlapping circles. This is the only microstructure out of the six that can be generated by the brute force method in O($S^2$), while the others are bounded roughly at O($S^3$) with brute force generation. Figure 11b shows a microstructure generated using non-overlapping circles with a prescribed minimum separation, while Figure 11c shows an example using circles that overlap by 5% to 25% of their area. Figure 11d–f are generated using shapes with meta-local states such as the ones shown in Figure 4. Figure 11d is made by ensuring wider angled overlap between tips of placed rectangles, while Figure 11e enforces overlap between the centers of already placed rectangles and tips of rectangles to be placed. Figure 11f is, first, generated using random non-overlapping circles; then, it is modified by placing rectangles with tip regions that overlap the existing circles. The variety seen in these structures demonstrates the versatility of the novel protocols presented in this work, even when only basic geometric shapes are used. With the utilization of richer object libraries, it should be possible to efficiently generate a very large variety of microstructures.

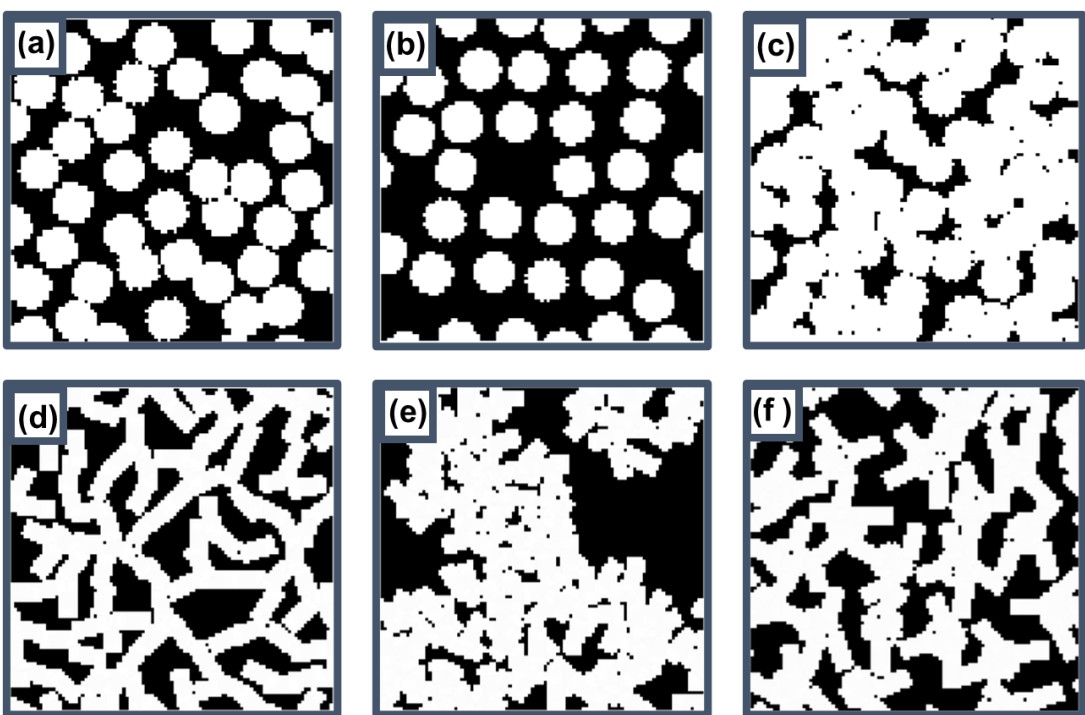

**Figure 11.** Various examples of digitally generated microstructures using only overlap and distance constraints: (**a**) non-overlapping circles, (**b**) non-overlapping circles with a prescribed minimum separation, (**c**) overlapping circles with 5% to 25% of their area, (**d**) wider angled overlap between tips of placed rectangles, (**e**) overlap between the centers of already placed rectangles and tips of rectangles to be placed, and (**f**) random non-overlapping circles modified by placing rectangles with tip regions that overlap the existing circles.

### 3.2. Generation of Statistically Similar Microstructures

As the final example, we demonstrate microstructure generation under two-point statistical guidance. Figure 12a shows a segmented experimental image obtained by scanning electron microscopy (SEM). The task is to generate additional microstructure instances that could have come from the same population this sample was taken. The information from the experimental image sample is used in two ways: (i) the two-point statistics of the experimental image (see Figure 12b) are used as the target statistics, and (ii) an object library consisting of 36 objects was manually mined from this image (see Figure 13). Notice that, while there are objects of various size and shapes in the library, all objects predominantly contain rounded features.

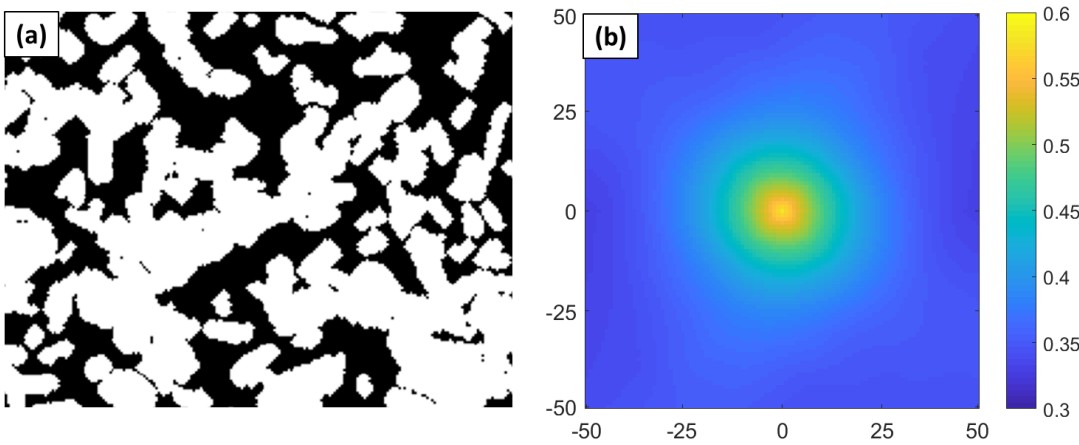

**Figure 12.** (**a**) Segmented version of a microstructural image acquired via scanning electron microscope. (**b**) Corresponding autocorrelation (trimmed to show short range vectors) of the segmented microstructure.

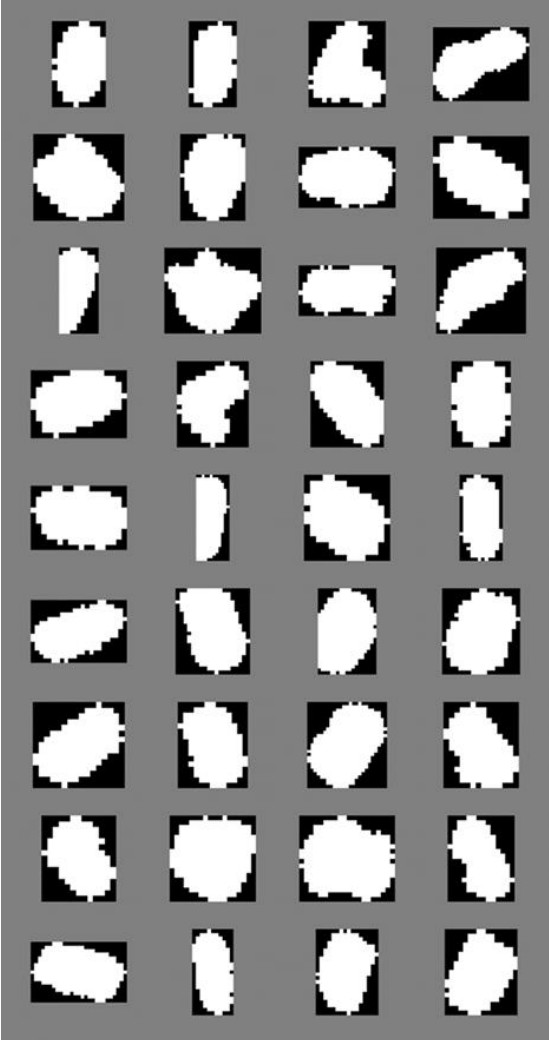

**Figure 13.** The object library mined from Figure 12 for the generation of statistically similar microstructures.

We approach the microstructure generation procedure in the following steps:

(1)   Start with a blank canvas.
(2)   While the volume fraction of the current image is less than the target image:

 a    Sample an object randomly from the object library
 b    Randomly rotate the object
 c    Compute overlap cost functions with 0% to 50% overlap range, as suggested by a visual inspection of the SEM image.
 d    Compute statistical cost functions, set the statistics of the segmented SEM image as the target statistics.
 e    Multiplicatively join the OCFs and SCFs.
 f    Sample a point to place the object assuming the resulting cost function represents a probability distribution (with proper normalization).

Figure 14 shows three example microstructures generated using the procedure described earlier. Notice that, while looking significantly different overall, all three generated microstructures share many common features with the original microstructure (see Figure 12a), including but not limited to: (i) the presence and frequency of small island-like formations, (ii) the presence of major directional chains composed of many overlapping objects, (iii) occasional large gaps between features, and (iv) a high degree of similarity between their two-point statistics (see Figures 12 and 14).

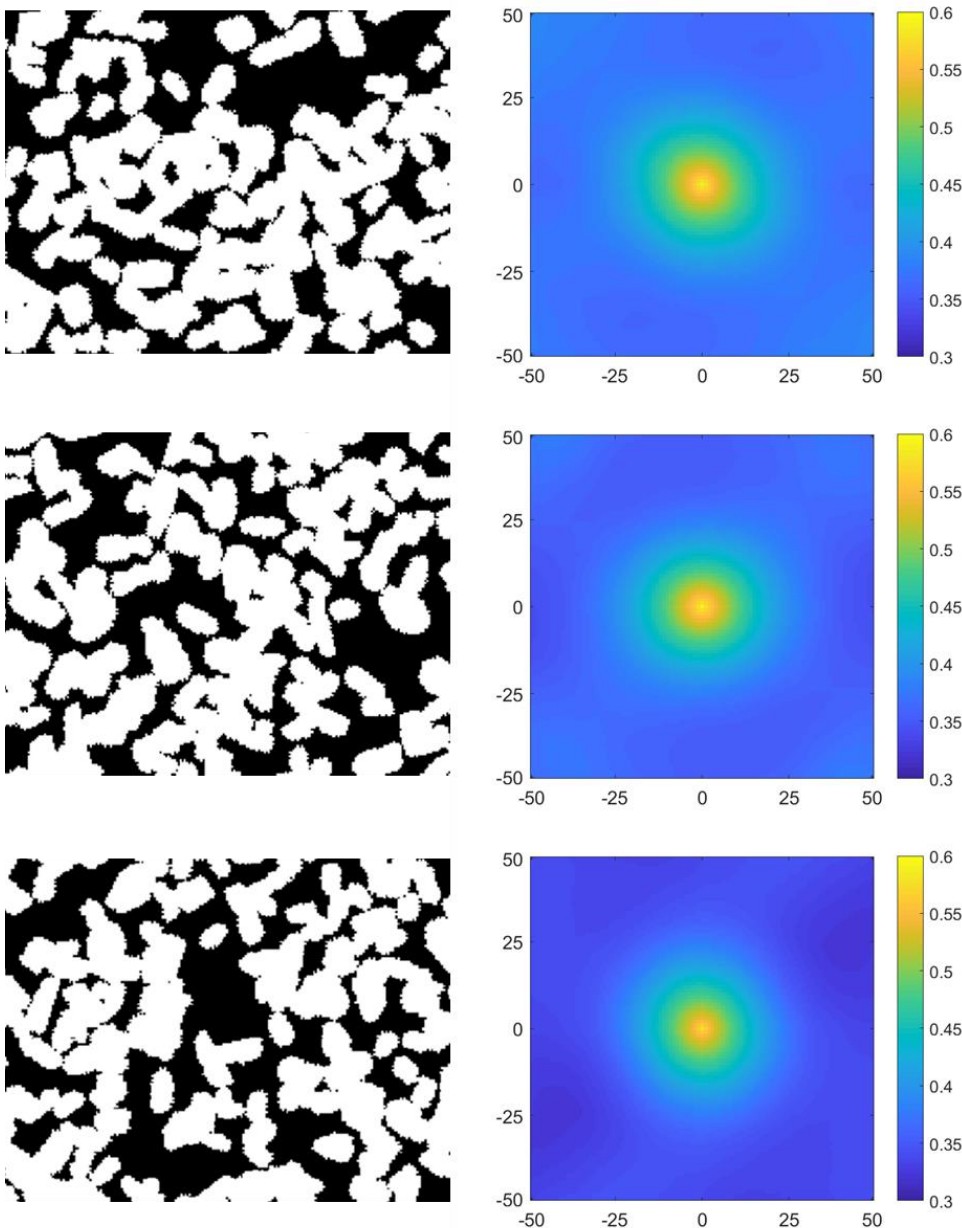

**Figure 14.** Three examples of microstructures generated using the mined object library using statistical cost functions.

## 4. Conclusions

A framework enabling rapid and scalable generation of microstructures under geometrical and statistical constraints has been developed and demonstrated with several examples. The new framework enables users to generate ensembles of digital microstructures in a computationally efficient manner by allowing users to define statistical constraints as well as geometric constraints in flexible but modular workflows. In addition, the new framework enables users to generate digital microstructures that are statistically similar to those documented in experiments by utilizing arbitrary shaped features mined directly from the experimental micrographs. The examples showcase the versatility of the proposed microstructure generation framework. The framework accomplishes the needed computations with only O(SlogS) computational complexity. The framework allows the insertion of numerous user-defined hyper-parameters and the exploration of multiple optimization alternatives in order to generate a large and diverse library of composite microstructures.

This framework is offered as a foundational step in the design and development of materials with heterogeneous (i.e., statistically homogeneous) microstructures.

**Author Contributions:** Conceptualization, A.C.; methodology, A.C. and B.Y.; software, A.C. and B.Y.; validation, A.C., B.Y. and S.R.K.; writing—original draft preparation, A.C.; writing—review and editing, B.Y and S.R.K.; supervision, S.R.K.; project administration, S.R.K.; funding acquisition, S.R.K. All authors have read and agreed to the published version of the manuscript.

**Funding:** This research was funded by National Science Foundation, grant number 1761406.

**Institutional Review Board Statement:** Not applicable.

**Informed Consent Statement:** Not applicable.

**Conflicts of Interest:** The authors declare no conflict of interest.

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
