# Peer review of "A Generalized and Modular Framework for Digital Generation of Composite Microstructures"

_jcs, doi:10.3390/jcs5080211_

Round 1
Reviewer 1 Report
The manuscript "A generalized and modular framework for digital generation of composite microstructures" is an original and a very interesting-to-the-reader work. I think the quality of some parts can be improved, such as Conclusions, but overall this is a good work and I recommend it for publication. See attached commented manuscript
//Accept

Author Response
I think the quality of some parts can be improved, such as Conclusions, but overall this is a good work and I recommend it for publication.
Author Response: We thank you for the comment. We updated the conclusion section as suggested. Please see the page 20 lines 558-563.
Reviewer 2 Report
Comments
This paper studied a generalized and modular framework for digital generation of composite microstructures. However, there are several aspects that need to be improved. The reviewer can only recommend for publication if the author satisfactorily address the following major comments in the revised version.
- The research questions and justification of selected parameters should be highlighted.
- The results and discussion was written in general. It is expected that the effect of key parameters should be discussed.
- The novelty of the study should be highlighted more clearly at the end of introduction section. How this study is different from the published study in literature?
- How the outcome of this study will benefit researchers and end users? This need to be highlighted in introduction or end of conclusion.
- The importance of composite microstructures and the recent investigation should be discussed in introduction section to improve the background study. Recently, the microstructure of composites [Ref: Testing and modelling the fatigue behaviour of GFRP composites–Effect of stress level, stress concentration and frequency] and the effect of additives of microstructural properties of composites [Ref: Effect of fire-retardant ceram powder on the properties of phenolic-based GFRP composites] are discussed. Suggest to include them in introduction section with proper citations to improve the background study.
I would be happy to see the revised version to understand how these comments are being addressed.
Author Response
The research questions and justification of selected parameters should be highlighted.
Author Response: We thank you for the comment. The research question was highlighted in the introduction section on page 3, lines 114-150. We discussed that there is a need for a new microstructure generation framework that can handle multiple expressions of geometrical and statistical constraints (in a computationally efficient manner). We propose an overarching framework that provides tremendous flexibility in formulating the cost functions. The selected parameters are chosen randomly, to illustrate the versatility of the novel protocols presented in this work.
The results and discussion was written in general. It is expected that the effect of key parameters should be discussed.
Author Response: They key parameters for the novel microstructure generation framework are the target overlap constraint, target distance constraint and the target two-point stats map. The effects of the overlap constraint and distance constraint on the placement was demonstrated in the Figure 3 where we showed the various placement modes that can be achieved using different overlap and distance constraints. We also demonstrated in Figure 11 various examples of digitally generated microstructures using different overlap and distance constraints. Also, the effect of the target two-point statistics map was demonstrated in the final case study (section 3.2 Generation of Statistically similar Microstructures), where we emphasized the necessity of the SCF for generating statistically similar microstructures.
We highlighted the user-defined parameters in section 2 (Microstructure Generation Framework). Please see page 4-5 lines 190-195
The novelty of the study should be highlighted more clearly at the end of introduction section. How this study is different from the published study in literature?
Author Response: We discussed the deficiency of the existing microstructure generation techniques in the introduction section (page 2 line 80-88). The existing microstructure frameworks are not extended to situations where one desires to target a large number of different geometric and statistical constraints. Also, current approaches are not able to reproduce the complex features observed in experimentally documented microstructures. This new framework can accommodate both geometrical constraints (i.e., target overlap and distance constraint) and statistical constraints (i.e., target two point correlations map) which enables users to generate a wide range of microstructures exhibiting desired characteristics. The framework also allows specification of arbitrary shaped features (could come directly from experimental microstructures) that enables users to reproduce the complex features observed in experimentally documented microstructures (demonstrated in section 3.2). This discussion is in the last paragraph of the introduction (page 3 line 141-150)
How the outcome of this study will benefit researchers and end users? This need to be highlighted in introduction or end of conclusion.
Author Response: As discussed in the introduction section, the new framework enables users to generate a wide variety of digital microstructures meeting their specifications, which can be expressed in multiple ways. Moreover, the computations involved are carried out at O(S) complexity, which is significantly better than any of the other methods currently used in literature.
The conclusion section is updated to address reviewer’s commend ( please see the page 20 lines 558-563)
The importance of composite microstructures and the recent investigation should be discussed in introduction section to improve the background study. Recently, the microstructure of composites [Ref: Testing and modelling the fatigue behaviour of GFRP composites–Effect of stress level, stress concentration and frequency] and the effect of additives of microstructural properties of composites [Ref: Effect of fire-retardant ceram powder on the properties of phenolic-based GFRP composites] are discussed. Suggest to include them in introduction section with proper citations to improve the background study.
Author Response: Suggested citations were added to the revised manuscript.
Reviewer 3 Report
The research has been dedicated to present a new approach title filter-based scheme to generate a composite microstructure. The study was interesting, however, it is not complete and well-organized. According to the following comments, it cannot be recommended for publication in this journal. 1- The main contribution of this study is not clearly presented. 2- The literature review of the study is missing. 3- The validation study has not been done. So the results are not reliable. 4- Some details of computations and procedure are missing. 5- It is not clear how the authors investigate the periodic boundary condition and the intersection of particle. 6- The author should provide a flowchart in which the detail of each step has been presented, such as an algorithm. 7- The authors have not discussed the maximum volume fraction of particles which can be considered by this method in the microstructure. 8- Most of the figures include several parts. It is better that the author assign a sub-number for each part like figure 2 (a) and figure 7, 10, 11 and etc. Also, for each part, required information should be discussed. 9- The conclusion part should be re-written. The main results only obtained from this study should be presented clearly at this part.Author Response
The main contribution of this study is not clearly presented.
Author Response: We discussed the deficiency of the existing microstructure generation techniques in the introduction section (page 2 line 80-88). The existing microstructure frameworks are not extended to situations where one desires to target a large number of different combinations of geometrical and statistical requirements. As a result, current approaches are not able to reproduce statisfactorily the complex features observed in experimentally documented microstructures. The new framework can accommodate an arbitrary combination of geometrical (expressed as target overlap and distance constraints) and statistical (i.e., two-point correlations) specifications, which offers users tremendous flexibility in generating digital microstructures for their work. The framework also allows specification of arbitrary shaped features (could come directly from experimental microstructures) that enables users to reproduce the complex features observed in experimentally documented microstructures (demonstrated in section 3.2). Most importantly, all of the computations can be performed at O(NlogN) complexity, which is far better than the competing approaches in current literature( page 6 line 264-270). This discussion is in the last paragraph of the introduction (page 3 line 141-150).
The literature review of the study is missing.
Author Response: We discussed the existing digital microstructure generation studies in the first three paragraph of the introduction section. We point out the recent efforts in the field of generation of the digital microstructures for various problem statements (citations 6-25 and 29-30).
The validation study has not been done. So the results are not reliable.
Author Response: The case studies shown in this study are the validation studies. They show that our protocols generate microstructures that successfully meet the user defined specifications (both statistical and geometrical). Also, they are done with O(NlogN) computational complexity. This computational efficiency also validates our claim that the filter-based approaches (discussed in this study) provide remarkable computational advantages compared to the conventional approaches used currently for microstructure generation. The results are very reliable and have been shown to meet the specifications.
Some details of computations and procedure are missing.
Author Response: All details of the computations for geometric (overlap and distance) cost functions were presented in equations 1-4 and all details of the computations for the statistical cost functions were presented in equations 5-6. We also discussed the details of the joint cost function (equation 7) in the section 2.3.
It is not clear how the authors investigate the periodic boundary condition and the intersection of particle.
Author Response: Equation 1 shows the calculations for OCFs where we use the FFT algorithm for the OCF computations. One of the consequences of employing DFTs is that they implicitly impose periodicity conditions across the image boundaries. We have this discussion at page 7 lines 288-299. We also referenced [44] for a description of the padding options for the non-periodic cases. There is no issue with the particle’s intersection with boundary in our computations, as the image is wrapped around the boundaries.
The author should provide a flowchart in which the detail of each step has been presented, such as an algorithm.
Author Response: The main workflow and the steps were presented in Figure 1. All the mathematical details of the steps have been presented in a separate section (named consistent with the steps depicted in Figure 1).
The authors have not discussed the maximum volume fraction of particles which can be considered by this method in the microstructure.
Author Response: We did not investigate the maximum volume fraction of particles which can be accommodated in the microstructure, as this was not the goal of our work.
Most of the figures include several parts. It is better that the author assign a sub-number for each part like figure 2 (a) and figure 7, 10, 11 and etc. Also, for each part, required information should be discussed.
Author Response: Figure numbering (i.e. (a), (b)) and captions are updated as suggested.
The conclusion part should be re-written. The main results only obtained from this study should be presented clearly at this part.
Author Response: We updated the conclusion section as suggested (please see the page 20 lines 558-563).
Round 2
Reviewer 2 Report
I have no further comments
Reviewer 3 Report
All comments have been successfully considered by the authors.